# The Impact of the Physical Activity Level on Sarcopenic Obesity in Community-Dwelling Older Adults

**DOI:** 10.3390/healthcare12030349

**Published:** 2024-01-30

**Authors:** Seongmin Choi, Jinmann Chon, Myung Chul Yoo, Ga Yang Shim, Minjung Kim, Miji Kim, Yunsoo Soh, Chang Won Won

**Affiliations:** 1Department of Physical Medicine and Rehabilitation Medicine, Kyung Hee University Hospital, Seoul 02447, Republic of Korea; 2Department of Physical Medicine and Rehabilitation, Graduate School, Kyung Hee University, Seoul 02447, Republic of Korea; 3Department of Health Sciences and Technology, College of Medicine, Kyung Hee University, Seoul 02447, Republic of Korea; 4Department of Family Medicine, College of Medicine, Kyung Hee University, Seoul 02447, Republic of Korea

**Keywords:** sarcopenic obesity, physical activity, aging

## Abstract

Previous studies have reported that low levels of physical activity result in sarcopenic obesity (SO). However, the effects of specific intensities of physical activity on SO and the optimal amount of physical activity for lowering the prevalence of SO have not been well studied. This study aimed to identify the effects of physical activity levels and intensity on SO and the optimal amount of physical activity related to a lower prevalence of SO. This cross-sectional study used data from the nationwide Korean Frailty and Aging Cohort Study (KFACS), which included 2071 older adults (1030 men, 1041 women). SO was defined according to the criteria of the European Society for Clinical Nutrition Metabolism (ESPEN) and the European Association for the Study of Obesity (EASO). Multivariate logistic regression analysis was performed to investigate the association between the physical activity level and SO. The high activity group had a significantly lower prevalence of SO than the non-high activity (low and moderate activity) group. On the other hand, moderate-intensity physical activity was associated with a lower prevalence of SO. A total physical activity energy expenditure of > 3032 kcal/week (433 kcal/day) for men and 2730 kcal/week (390 kcal/day) for women was associated with a reduced prevalence of SO. The high physical activity and total physical energy expenditure described above may be beneficial for reducing the prevalence of SO.

## 1. Introduction

Concomitant with socioeconomic development, the proportions of older adults and obese individuals have increased. Worldwide, the percentage of the population aged ≥65 years is expected to increase from 9.8% by 2022 to approximately 16.0% by 2050 [1]. Obesity in older adults is related to lifestyle changes, including low physical activity, high-calory eating habits, and physiological changes such as lower sex hormones [2]. In addition, vertebral body compression reduces height and increases body mass index (BMI), which is calculated as weight divided by height squared [3]. The prevalence of obesity among adults aged ≥70 years with a BMI of ≥25 kg/m^2^ has increased from 31.9% in 2009 to 35.5% in 2018, indicating a growing trend in South Korea [4]. Obesity causes metabolic syndrome in older adults, and weight gain due to fat accumulation induces insulin resistance and hyperinsulinemia in the liver and peripheral tissues. This could increase the risk of developing hypertension, glucose intolerance, diabetes, visceral obesity, dyslipidemia, cardiovascular disease, and some cancers [5]. The evaluation and prevention of obesity are essential for decreasing comorbidities and mortality in older adults.

However, after the fourth decade of life, the skeletal muscle mass decreases, altering the body weight and composition [6]. Age-related decreases in muscle mass cause sarcopenia, which is defined as a reduction in skeletal muscle mass, strength, and/or physical function [7]. Decreased total energy consumption, such as reduced resting metabolic rate and lower physical activity often seen during aging, contributes substantially to an increase in body fat mass [6]. Age-related changes in sex hormones such as testosterone and estrogen affect muscle mass and body fat. In men, testosterone induces muscle regeneration through satellite cell activation. However, testosterone levels decrease by 1–3% annually with age, which may contribute to muscle loss and fat distribution in older men [8]. In women, estrogen modulates inflammation and satellite cell activation in skeletal muscles. After menopause, body weight increases, especially visceral fat, which causes an increase in waist circumference [9]. Sarcopenia exacerbates the negative impacts of obesity in older individuals, leading to the development of sarcopenic obesity (SO). It is associated with a synergistic exacerbation effect of functional impairment, increased mortality, and reduced quality of life, leading to long-term disabilities in older adults. [10].

Physical activity decreases with age, and reports have suggested that the degree of physical activity in daily life affects the prevalence of sarcopenia and obesity, increasing the risk of developing SO [11]. Previous studies have reported that high physical activity levels are associated with a reduced risk of developing SO [12,13,14]. However, the definition of SO used in previous studies is inconsistent because of the lack of universally recognized diagnostic criteria. In addition, the effects of specific intensities of physical activity on SO, individual components of SO, and optimal amount of physical activity required to lower the risk of SO have not been well studied. Until recently, no standardized definition of SO has been available. Various operational definitions have been proposed to describe SO. In 2022, the European Society for Clinical Nutrition and Metabolism (ESPEN) and the European Association for the Study of Obesity (EASO) jointly established a diagnostic definition of the diagnostic criteria for SO [15]. The ESPEN and EASO criteria showed more reasonable and universally valid results because SO was determined by considering the differences in body composition according to race. In the current cross-sectional cohort analysis, we investigated the effects of intensity and the optimal amount of physical activity on SO, as defined by the ESPEN and EASO criteria, in a large group of community-dwelling older adults enrolled in the nationwide Korean Frailty and Aging Cohort Study (KFACS).

## 2. Materials and Methods

### 2.1. Study Population

This cross-sectional study used data from the KFACS collected between May 2016 and November 2017. The KFACS, a nationwide study, was conducted at ten centers comprised of eight medical hospitals and two public health centers. The KFACS recruited 3014 community-dwelling older adults aged 70–84 years. The survey included questionnaires, face-to-face interviews, health examinations, and laboratory tests conducted at each clinical center. Laboratory tests were taken at 08:00 after 8 h of fasting. The research investigators had been trained at Kyung Hee University Hospital by KFACS staff members every year to ensure standardized quality [16]. Of the 3014 participants, 2403 who underwent dual-energy X-ray absorptiometry (DXA) were included. Participants with incomplete data on physical function tests, dementia or cognitive impairment, hemiplegia, artificial joints, or other metallic objects in the appendicular body regions were excluded from the study (Figure 1).

This study included 2071 older adults (1030 men, 1041 women). The demographic data and medical history included age, sex, body mass index (BMI), waist circumference, presence of chronic diseases or comorbidities, smoking and alcohol status, and physical activity status. Participants who smoked more than one cigarette per week were defined as smokers, and those who drank alcohol more than once per week were defined as alcohol consumers.

The KFACS protocol received approval from the Institutional Review Board (IRB) of the Clinical Research Ethics Committee at the Medical Center (IRB number: 2015-12-103), and all participants gave written informed consent.

### 2.2. Sarcopenic Obesity

SO was defined according to the ESPEN and EASO guidelines [15]. Participants with high fat mass, low appendicular lean mass (ALM), and either low handgrip strength (HGS) or abnormal five-time sit-to-stand chair test results were diagnosed with SO. A cutoff value for each parameter was determined, corresponding to the Asian values suggested by the ESPEN and EASO. 

(1)Fat mass: DXA was used to measure body fat mass, and the body fat mass was adjusted to body weight (body fat mass/weight) (cut-off values, men: >29.7%, women: >37.2%) [17].(2)Appendicular lean mass: DXA was used to measure ALM, which was adjusted for body weight (ALM/weight, ALM/W) (cutoff values, men: <29.5%, women: <23.2%) [18].(3)Muscle strength: HGS was measured using a hand dynamometer (T.K.K.5401; Takei Scientific Instruments Co., Ltd., Tokyo, Japan). The participants were instructed to squeeze the handle with maximum effort, with the elbow extended in a standing position for 3 s. HGS was measured twice on both sides, and the maximum value was obtained in kilograms (cutoff values: men <28 kg, women <18 kg) [19].(4)Five-time sit-to-stand test. The five-time sit-to-stand test assesses the time it takes to rise five times from a seated position without utilizing the arms. The participants were instructed to stand up and sit down five times as quickly as possible. Time was measured to the nearest 0.01 s (cut-off value, ≥17 s) [20].

### 2.3. Physical Activity

The short form of the International Physical Activity Questionnaire (IPAQ) was used to assess the extent of physical activity [21]. The IPAQ comprises questions regarding the duration and frequency of three specific types of activities: walking, moderate-intensity activities, and vigorous-intensity activities. The total physical activity level was expressed as metabolic equivalents (METs) and calculated by multiplying the duration (min), frequency of physical activities (times per week), and MET for each type of activity [21].
Total physical activity [METs-min/week] = [3.3{METs} × min × days] + [4.0{METs} × min × days] + [8.0{METs} × min × days]

The participants were classified into three levels of physical activity based on the IPAQ guidelines, shown in Appendix A: high activity (any one of the following two criteria: 7 d of walking or moderate or vigorous intensity activities accumulating ≥3000 MET-min/week or vigorous intensity activity on ≥3 d and ≥1500 MET-min/week); moderate activity (any one of the following three criteria: ≥5 d of walking or moderate or vigorous intensity activities accumulating ≥600 MET-min/week, or ≥5 d of 30 min of moderate intensity and/or walking per day, or ≥3 d of 20 min of vigorous activity per day); and low activity (not enough to meet moderate or high activity criteria) [21,22]. The total energy expenditure (kilocalories) was calculated from MET-min using the following equation [21]:Total energy expenditure (kCal) = MET-min × (weight in kg/60 kg)

### 2.4. Statistical Analysis

Continuous variables were compared through Student’s t-test or the Mann–Whitney U test, while categorical variables were assessed using the Pearson chi-square test. Mean ± standard deviation represented continuous variables, while numbers and ratios (%) expressed categorical variables. Receiver operating characteristic (ROC) curve analysis identified the optimal cut-off value. Odds ratios (ORs) and 95% confidence intervals (CIs) were computed using unadjusted and fully adjusted logistic regression models. The analysis was adjusted for confounding variables, including age, height, hypertension, dyslipidemia, cerebrovascular accident, osteoarthritis, osteoporosis, diabetes mellitus, depression, alcohol history, smoking history, heart disease, and the Mini-Mental State Examination of the Korean version of the CERAD Assessment Packet (MMSE-KC) score. Statistical analyses were conducted using the Statistical Package for Social Sciences (version 25.0; SPSS Inc., Chicago, IL, USA), and statistical significance was defined as *p* < 0.05.

## 3. Results

The baseline characteristics of the study participants are presented in Table 1. Of the 2071 participants, 1030 (49.7%) were men, and 1041 (50.3%) were women. The prevalence of SO was 8.7% in men and 10.4% in women. The total physical activity (METs-min/week) was significantly higher in the non-SO group than in the SO group. Other characteristics such as age, height, weight, BMI, waist circumference, and the prevalence of hypertension and diabetes mellitus were significantly higher in the SO group.

Table 2 shows the results of the logistic regression analysis of the association between physical activity and SO. In the multivariate analysis, the high activity group had a significantly lower prevalence of SO (odds ratio [OR], 0.53; 95% confidence interval [CI], 0.33–0.87 in men; OR, 0.39; 95% CI, 0.24–0.62 in women) than the non-high activity (low and moderate activity) group. Regarding the effect of the specific intensity of physical activity, a one hour increase in moderate-intensity physical activity was associated with a lower likelihood of SO (4.0 MET, OR, 0.68; 95% CI, 0.51–0.91 in men; OR, 0.83, 95% CI, 0.70–0.98 in women and 3.3 MET; OR, 0.74; 95% CI, 0.58–0.95 in men; OR, 0.66; 95% CI, 0.51–0.87 in women). 

The results of the logistic regression analysis between physical activity and individual SO components are presented in Table 3. In the multivariate analysis, participants in the high activity group showed a lower prevalence of high fat mass (OR, 0.65; 95% CI, 0.50–0.84 in women), low muscle mass (OR, 0.67; 95% CI, 0.52–0.87 in men; OR, 0.51; 95% CI, 0.39–0.66 in women), and low skeletal muscle function (OR, 0.63; 95% CI, 0.45–0.87 in men; OR, 0.44; 95% CI, 0.32–0.60 in women) than those in the non-high activity (low and moderate activity) group.

Table 4 shows the results of the logistic regression analysis between physical activity energy expenditure and SO and its components. In the multivariate analysis, the high energy expenditure group, defined by a cut-off value, had a significantly lower prevalence of SO (OR, 0.44; 95% CI, 0.27–0.71 in men; OR, 0.43; 95% CI, 0.27–0.67 in women), low muscle mass (OR, 0.76; 95% CI, 0.59–0.99 in men; OR, 0.58; 95% CI, 0.44–0.75 in women), and low skeletal muscle function (OR, 0.52; 95% CI, 0.37–0.72 in men; OR, 0.40; 95% CI, 0.29–0.55 in women), compared with the non-high energy expenditure group (cut-off value: 3032 kcal/week (433 kcal/day) for men and 2730 kcal/week (390 kcal/day) for women).

## 4. Discussion

This study investigated the association between physical activity and SO and its individual components. A high physical activity level was associated with a lower prevalence of SO. This may be due to the positive effects of high physical activity levels on the individual components of SO. High physical activity was associated with a lower prevalence of low muscle mass and skeletal muscle function in both sexes and high fat mass in women. Regarding the intensity and optimal amount of physical activity expenditure required to reduce the prevalence of SO, moderate-intensity exercise was associated with a lower prevalence. Total physical activity energy expenditures of ≥3032 kcal/week (433 kcal/day) for men and ≥2730 kcal/week (390 kcal/day) for women were associated with a reduced prevalence of SO.

Regular physical activity helps with weight loss, improves physical function, and slows the muscle loss associated with aging [23]. Although the benefits of physical activity on obesity or sarcopenia in older adults have generally been reported, studies on the effects of physical activity on SO are limited [24,25,26]. Further investigations are necessary to determine the effect of physical activity on SO as a unique clinical condition. Because a bidirectional pathogenic interaction exists between loss of muscle mass, function, and fat mass accumulation, SO has a synergistically higher risk for functional impairment and metabolic disease, compared with the cumulative risk of each illness [10,15,27]. 

Consistent with our results, several previous studies have reported the potential benefits of physical activity on SO in older adults. A cross-sectional study in Korea of 2264 older adults aged ≥65 y showed that increased physical activity, as assessed using IPAQ, was associated with a lower risk of developing SO. Participating in moderate (≥ 600 MET-min/week) and high (≥ 3000 MET-min/week) levels of physical activity significantly lowered the risk of developing SO by 51% and 75%, respectively [14]. Aggio et al. conducted a cross-sectional study to assess the association between objectively measured physical activity and SO in older British men and found that an additional 30 min of moderate-to-vigorous intensity physical activity per day was associated with a reduced risk of SO. They also found that sedentary time was associated with an increased risk of SO, independent of physical activity [13]. In addition, a multinational study using data from the Collaborative Research on Aging in Europe, the WHO Health Organization Study on Global Aging, and adult health surveys performed in nine countries reported that lower physical activity based on the Global Physical Activity Questionnaire was strongly associated with a higher risk of SO. Individuals with SO had lower physical activity levels than those with sarcopenia alone [28]. Although these studies have consistently reported the potential positive effects of physical activity on SO, the optimal intensity and total amount of physical activity required to lower the risk of SO have not been determined.

Receiver operating characteristic (ROC) analysis was performed to determine the optimal amount of physical activity required to reduce the risk of developing SO. Based on the ROC analysis, the optimal total physical energy expenditures to reduce the risk of SO was determined to be ≥3032 kcal/week (433 kcal/day) in men and ≥2730 kcal/week (390 kcal/day) in women. Studies have been conducted on the amount of physical energy expenditure required to reduce the risk of chronic diseases, such as cerebrovascular accidents, and maintain a healthy body weight. According to these studies, 150 min/week of moderate-intensity physical activity was associated with cerebrovascular health benefits, and 150–250 min/week (1200–2000 kcal/week) of moderately-vigorous physical activity was required to prevent weight gain. In addition, 225–420 min/week (2000 to 3700 kcal/day) of moderate-intensity physical activity resulted in weight loss of 5–7.5 kg, and 200–300 min/week of moderate physical activity was needed for weight maintenance after weight loss [29,30]. Although SO is defined by fat mass, muscle mass, and skeletal muscle function, the optimal total physical energy expenditure for lowering the risk of SO in the current study was comparable to the energy expenditure required for losing body weight or maintaining weight after weight loss. To achieve the energy expenditure for reducing the risk of SO reported in our results, approximately 90 min of moderate-intensity physical activity (3–4 MET), such as walking, carrying light loads, parenting, or cleaning almost all days of the week, may be needed.

Moderate-intensity physical activity is associated with a lower prevalence of developing SO. In contrast, vigorous-intensity activity was not associated with SO. The benefits of vigorous-intensity physical activity on cardiorespiratory fitness have been previously reported [31,32]. Despite its shorter duration and lower energy expenditure, vigorous-intensity training induced greater improvements in cardiorespiratory fitness than moderate physical activity [33]. However, vigorous-intensity physical activity did not affect body composition [34]. Moreover, in a meta-analysis, Pattyn et al. showed a more pronounced effect of moderate-intensity physical activity than vigorous physical activity on body composition [35]. These findings may be attributed to several factors. High-intensity physical activity can only be sustained for a relatively short period, leading to a lower energy expenditure than moderate physical activity. Additionally, moderate physical activity may activate fat metabolism more than a greater reliance on high-intensity carbohydrates [35].

This study has several limitations. Due to the cross-sectional study design, causal relationships between physical activity and SO could not be determined. Second, physical activity was measured based on self-reports, which may not accurately reflect energy expenditure. Because the IPAQ is a retrospective questionnaire, it is subject to bias. For example, recall bias, social desirability bias, and subjective interpretation may make it difficult to assess social activity accurately. Longitudinal studies using objective measurements of physical activity may demonstrate a considerable effect of physical activity on the risk of SO. Third, the nutritional status of participants, which may have affected their body composition, was not considered. This cross-sectional study used data between 2016 and 2017; so, seasonal and yearly variable limitations could exist. Finally, this study was conducted in a single race, and the results may not apply to other races. Despite these limitations, our study has the strength of presenting a specific cutoff kcal and intensity of physical activity associated with a lower prevalence of SO.

## 5. Conclusions

We found that a high physical activity level was associated with a lower prevalence of SO. It was also associated with a lower prevalence of low muscle mass and poor skeletal muscle function, both of which are included in the diagnostic criteria for SO. Moderate-intensity physical activity may be beneficial in reducing the prevalence of SO. Total physical energy expenditures of ≥3032 kcal/week (433 kcal/day) in men and ≥2730 kcal/week (390 kcal/day) in women may help reduce the prevalence of SO. Our findings suggest an appropriate intensity and amount of physical activity to lower the prevalence of SO.

## Figures and Tables

**Figure 1 healthcare-12-00349-f001:**
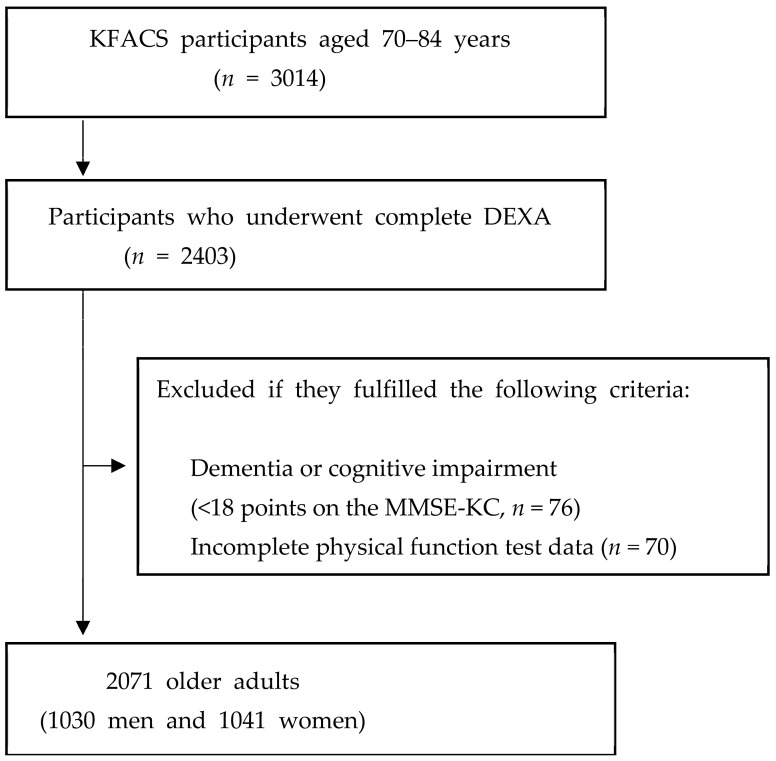
Flowchart of the participant recruitment process. KFACS, Korean Frailty and Aging Cohort Study; DEXA, Dual-energy X-ray absorptiometry; MMSE-KC, Mini-Mental State Examination of the Korean version of the CERAD Assessment Packet.

**Figure 2 healthcare-12-00349-f002:**
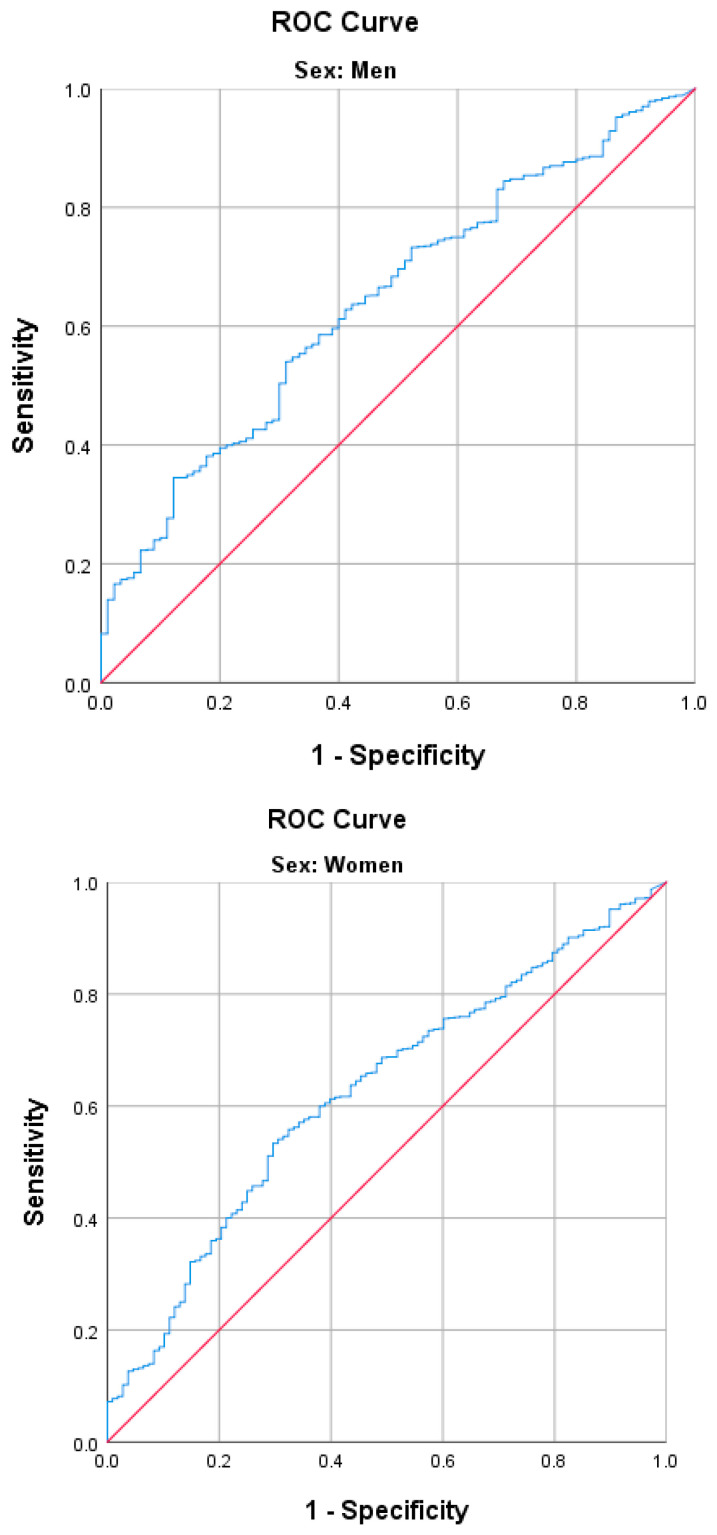
Receiver operating characteristic (ROC) curve analysis of physical activity energy expenditure for sarcopenic obesity. AUC for men, 0.625 cut-off value, 3032 kcal/week (54.0% sensitivity and 68.9% specificity, (95% CI, 0.57–0.68)); AUC for women, 0.643 cut-off value, 2730 kcal/week (53.3% sensitivity and 70.4% specificity 95% (CI, 0.59–0.70)).

**Table 1 healthcare-12-00349-t001:** Baseline characteristics of the participants according to sex.

	Men (*n* = 1030)		Women (*n* = 1041)	
	SO † (+) (*n* = 90, 8.7%)	SO (−) (*n* = 940, 91.3%)	SO † (+) (*n* = 108, 10.4%)	SO (−) (*n* = 933, 89.6%)
Age (years)	78.7 ± 3.9 **	76.1 ± 3.88	76.6 ± 3.9 **	75.3 ± 3.9
Height (cm)	163 ± 5.8 **	165 ± 5.5	150 ± 5.3 **	152 ± 5.2
Weight (kg)	67.8 ± 9.5 **	64.9 ± 8.8	58.6 ± 7.6 **	56.2 ± 7.6
BMI (kg/m^2^)	25.6 ± 2.6 **	23.7 ± 2.8	25.8 ± 2.7 **	24.2 ± 2.8
Waist circumference (cm)	94.8 ± 7.8 **	88.0 ± 8.2	89.7 ± 8.4 **	85.6 ± 8.0
Hypertension (*n*, %)	60 (66.7) **	485 (51.6)	74 (68.5) *	528 (56.6)
Diabetes mellitus (*n*, %)	32 (35.6) **	220 (23.4)	26 (24.1)	173 (18.5)
Dyslipidemia (*n*, %)	27 (30.0)	223 (23.7)	52 (44.1)	373 (40.0)
Heart disease (*n*, %)	11 (12.2)	88 (9.4)	7 (6.5)	55 (5.9)
CVA (*n*, %)	4 (4.4)	56 (6.0)	4 (3.7)	28 (3.0)
Alcohol (*n*, %)	46 (47.8)	483 (51.4)	13 (11.6)	99 (10.6)
Current smoker (*n*, %)	8 (9.8)	104 (11.1)	1 (0.9)	10 (1.1)
Knee OA (*n*, %)	12 (13.3)	97 (10.3)	33 (30.6)	267 (28.6)
Osteoporosis (*n*, %)	4 (4.4)	25 (2.7)	28 (25.9)	225 (24.1)
Depression (*n*, %)	4 (4.4)	15 (1.6)	8 (7.4) *	26 (2.8)
MMSE-KC	26.11 ± 3.01	26.43 ± 2.67	24.4 ± 3.8 **	25.4 ± 3.3
Total physical activity (METs-min/week) ¶	2320 ± 1833 **	4319 ± 4549	2606 ± 2242 **	4158 ± 3822
Total energy expenditure (Kcal/week) §	2641 ± 2180 **	4635 ± 5026	2526 ± 2224 **	3851 ± 3371
Biochemical variables				
Fasting glucose (mg/dL)	109.7 ± 21.7	104.8 ± 24.1	103.7 ± 30.8	102.2 ± 21.1
HbA1c (%)	6.1 ± 0.7 *	5.9 ± 0.8	6.1 ± 1.1	6.03 ± 0.76
Total cholesterol (mg/dL)	169.7 ± 33.6	168.1 ± 34.9	185.0 ± 40.3	181.4 ± 35.4
HDL-C (mg/dL)	46.0 ± 13.3 **	51.0 ± 14.3	57.3 ± 15.8	54.8 ± 13.5
LDL-C (mg/dL)	105.8 ± 30.3	104.1 ± 31.8	112.3 ± 35.2	113.1 ± 33.9
Protein (mg/dL)	7.0 ± 0.4	7.0 ± 0.4	6.9 ± 0.4	7.0 ± 0.3
Albumin (mg/dL)	4.3 ± 0.2	4.3 ± 0.2	4.3 ± 0.2	4.3 ± 0.2

BMI, body mass index; CVA, cerebrovascular accident; OA, osteoarthritis; MMSE-KC, Mini-Mental Status Examination—Korean version; MET, metabolic equivalent task; HDL-C, high-density lipoprotein cholesterol; LDL-C, low-density lipoprotein cholesterol. † SO, Sarcopenic obesity: high fat mass (>29.7% for men and >37.2% for women) and low appendicular lean mass/weight measured using DXA (<29.5% for men and <23.2% for women) and either low hand grip strength (<28 kg for men and <18 kg for women) or abnormal five-time sit-to-stand chair test (≥17 s). ¶ Total physical activity (METs-min/week × min × days): (3.3[METs] × min × days) + (4.0[METs] × min × days) + (8.0(METs] × min × days). § Total energy expenditure (kcal/week): total physical activity (METs-min/week) * (body weight [kg]/60). * *p* < 0.05. ** *p* < 0.01.

**Table 2 healthcare-12-00349-t002:** Logistic regression analysis of the physical activity levels and intensity categories predicting sarcopenic obesity.

	Unadjusted Model		Fully Adjusted Model	
	Men	Women	Men	Women
	OR (95% CI)	OR (95% CI)	OR (95% CI)	OR (95% CI)
Physical activity level				
Moderate Activity †	0.77 (0.40–1.49)	0.83 (0.46–1.48)	0.89 (0.44–1.77)	0.92 (0.50–1.70)
High activity ††	0.47 (0.29–0.74) *	0.35 (0.22–0.55) **	0.53 (0.33–0.87) *	0.39 (0.24–0.62) **
Physical intensity category				
8.0 MET ¶¶ (vigorous)	0.40 (0.16–0.98) *	0.39 (0.07–2.24)	0.46 (0.19–1.13)	0.56 (0.12–2.74)
4.0 MET ¶¶ (moderate)	0.61 (0.46–0.82) *	0.77 (0.64–0.92) *	0.68 (0.51–0.91) *	0.83 (0.70–0.98) *
3.3 MET ¶¶ (walking)	0.73 (0.57–0.93) *	0.62 (0.47–0.82)	0.74 (0.58–0.95) *	0.66 (0.51–0.87) *

Abbreviations: OR, odds ratio; CI, confidence interval. The fully adjusted model was adjusted for age, height, hypertension, dyslipidemia, cerebrovascular accident, osteoarthritis, osteoporosis, diabetes mellitus, depression, alcohol history, smoking history, heart disease, and the MMSE-KC score. Additionally, the physical activity category was adjusted for the other two intensities of physical activity. † Low activity as reference. †† Non-high activity including low and moderate activity as reference. High activity, any one of the following two criteria: 7 d of walking or moderate or vigorous intensity activities accumulating ≥3000 MET-min/week or vigorous intensity activity on ≥3 d and accumulating ≥1500 MET-min/week. Moderate activity, any of the following three criteria: ≥5 d of walking or moderate or vigorous intensity activities accumulating ≥600 MET-min/week or ≥5 d of 30 min of moderate intensity or walking per day or ≥3 d of 20 min of vigorous activity per day. Low activity is not sufficient to meet moderate or high activity criteria. A logistic regression analysis was performed based on the hours of activity for each exercise intensity per day. ¶¶ 8.0 MET (vigorous intensity activity), heavy lifting (>20 kg), digging or carrying things on the stairs; 4.0 MET (moderate intensity activity), carrying light loads, parenting or cleaning; 3.3 MET, walking. * *p* < 0.05. ** *p* < 0.01.

**Table 3 healthcare-12-00349-t003:** Logistic regression analysis of high physical activity predicting individual components of sarcopenic obesity.

	Unadjusted Model		Fully Adjusted Model	
	Men	Women	Men	Women
	OR (95% CI)	OR (95% CI)	OR (95% CI)	OR (95% CI)
High fat mass ¶	0.73 (0.56–0.96) *	0.68 (0.53–0.87) *	0.78 (0.59–1.04)	0.65 (0.50–0.84) *
Low muscle mass ¶	0.62 (0.49–0.80) **	0.53 (0.41–0.68) **	0.67 (0.52–0.87) *	0.51 (0.39–0.66) *
Low skeletal muscle function ¶	0.55 (0.41–0.74) **	0.39 (0.28–0.52) **	0.63 (0.45–0.87) *	0.44 (0.32–0.60) *

Abbreviations: OR, odds ratio; CI, confidence interval. The fully adjusted model was adjusted for age, height, hypertension, dyslipidemia, cerebrovascular accident, osteoarthritis, osteoporosis, diabetes mellitus, depression, alcohol history, smoking history, heart disease, and the MMSE-KC score. Non-high physical activity including low activity and moderate activity as reference. High activity, any one of the following 2 criteria: 7 days of walking or moderate or vigorous intensity activities accumulating ≥3000 MET-min/week or vigorous intensity activity on ≥3 days and accumulating ≥1500 MET-min/week. Moderate activity, any of the following 3 criteria: ≥5 days of walking or moderate or vigorous intensity activities accumulating ≥600 MET-min/week, or ≥5 days of 30 min of moderate intensity or walking per day, or ≥3 days of 20 min of vigorous activity per day. Low activity is not sufficient to meet moderate or high activity criteria. ¶ High fat mass, >29.7% for men and >37.2% for women. Low muscle mass and low appendicular lean mass/weight were observed in <29.5% of men and <23.2% of women. Low skeletal muscle function, either low hand grip strength (<28 kg for men and <18 kg for women) or abnormal five-times sit-to-stand chair test (≥17 s). * *p* < 0.05. ** *p* < 0.01. ROC analysis to determine the most appropriate cut-off value of physical activity energy expenditure for SO revealed that energy expenditures of ≥3032 kcal/week (433 kcal/day) in men and ≥2730 kcal/week (390 kcal/day) in women yielded optimal results (Figure 2).

**Table 4 healthcare-12-00349-t004:** Logistic regression analysis of high energy expenditure § predicting sarcopenic obesity and individual components of sarcopenic obesity.

	**Unadjusted Model**		**Fully Adjusted Model**	
	**Men**	**Women**	**Men**	**Women**
	**OR (95% CI)**	**OR (95% CI)**	**OR (95% CI)**	**OR (95% CI)**
Sarcopenic Obesity †	0.39 (0.24–0.61) **	0.37 (0.24–0.57) **	0.44 (0.27–0.71) *	0.43 (0.27–0.67) *
High fat mass ¶	0.88 (0.67–1.14)	0.87 (0.68–1.11)	0.90 (0.68–1.19)	0.81 (0.62–1.05)
Low muscle mass ¶	0.73 (0.57–0.93) *	0.61 (0.48–0.78) **	0.76 (0.59–0.99) *	0.58 (0.44–0.75) *
Low skeletal muscle function ¶	0.44 (0.33–0.59) **	0.33 (0.25–0.45) **	0.52 (0.37–0.72) *	0.40 (0.29–0.55) *

Abbreviations: OR, odds ratio; CI, confidence interval. The fully adjusted model was adjusted for age, height, hypertension, dyslipidemia, cerebrovascular accident, osteoarthritis, osteoporosis, diabetes mellitus, depression, alcohol history, smoking history, heart disease, and the MMSE-KC score. Non-high energy expenditure as reference. § Cut-off value: 3032 kcal/week (433 kcal/day) for men and 2730 kcal/week (390 kcal/day) for women, calculated using the receiver operating characteristic (ROC) curve. † Sarcopenic obesity: high fat mass (>29.7% for men and >37.2% for women) and low appendicular lean mass/weight measured using DEXA (<29.5% for men and <23.2% for women) and either low hand grip strength (<28 kg for men and <18 kg for women) or abnormal five-time sit-to-stand chair test (≥17 s). ¶ High fat mass, >29.7% for men and >37.2% for women. Low muscle mass and low appendicular lean mass/weight were observed in <29.5% of men and <23.2% of women. Low skeletal muscle function, either low hand grip strength (<28 kg for men and <18 kg for women) or abnormal five-time sit-to-stand chair test (≥17 s). * *p* < 0.05. ** *p* < 0.01.

## Data Availability

All cohort data supporting the findings of this study are available from the KFACS and are open to all researchers upon reasonable request. All news articles published in the KFACS database, data provision manuals, and contact information are available on the KFACS website (http://www.kfacs.kr).

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
