# Peer review of "The Impact of the Physical Activity Level on Sarcopenic Obesity in Community-Dwelling Older Adults"

_healthcare, 2024, doi:10.3390/healthcare12030349_

Round 1

Reviewer 1 Report

Comments and Suggestions for Authors

The research is relevant and necessary, as the authors rightly point out that sarcopenia and obesity in older adults are problems that will escalate over time. The strength of the study lies in the sample size.

The paper's structure is correct, indicating the authors' mastery of research methodology; from this perspective, it is impeccable. Therefore, I will recommend the publication of the paper, but in my opinion, some small modifications would enhance readability.

While the objective is perfectly clear in the abstract, at the end of the introduction (Lines 74-76), although the authors' intentions are understood, I believe the objective should be stated with greater precision.

The data cover the year 2016-2016, and although in this type of study, it is a non-determining variable, it should be acknowledged in the discussion as a limitation due to potential biases introduced in the results.

To assess physical activity levels, the IPAQ was used, and as the authors are well aware, the IPAQ introduces significant bias in evaluating physical activity levels. Therefore, this should be explicitly mentioned in the study's limitations. While the authors touch on this in Lines 356-358, I suggest specifically referencing the IPAQ and citing relevant literature, as there are many studies correlating the IPAQ with accelerometer measurements.

Although not critical, I recommend that, in Table 1, the authors remove p-values and replace them with asterisks for mean or percentage values. It's important to note that specialists will be reading the paper. Additionally, for most variables, indicating one decimal place is more than sufficient; excessive precision is not required.

Similarly, in Tables 2, 3, and 4, it is unnecessary to include p-values. It is well known that when the 95% CI does not contain zero, it is always statistically significant. While not obligatory, the authors could use only an asterisk to indicate whether the odds ratio is statistically significant.

Author Response

Reviewer 1

The research is relevant and necessary, as the authors rightly point out that sarcopenia and obesity in older adults are problems that will escalate over time. The strength of the study lies in the sample size. The paper's structure is correct, indicating the authors' mastery of research methodology; from this perspective, it is impeccable. Therefore, I will recommend the publication of the paper, but in my opinion, some small modifications would enhance readability.

While the objective is perfectly clear in the abstract, at the end of the introduction (Lines 74-76), although the authors' intentions are understood, I believe the objective should be stated with greater precision.

Response: Thank you for your thoughtful suggestion. We agree with you and have rerevised this content on lines 74-77.

In the current cross-sectional cohort analysis, we investigated the effects of intensity and the optimal amount of physical activity on SO as defined by the ESPEN and EASO cri-teria, in a large group of community-dwelling, older adults enrolled in the nationwide Korean Frailty and Aging Cohort Study (KFACS).

The data cover the year 2016-2017. Although in this type of study, it is a non-determining variable, it should be acknowledged in the discussion as a limitation due to potential biases introduced in the results.

Response: Thank you for your thoughtful suggestion. We added these comments in Lines 372-373.

This cross-sectional study used data between 2016 and 2017, so seasonal and yearly variable limitations could exist.

To assess physical activity levels, the IPAQ was used, and as the authors are well aware, the IPAQ introduces significant bias in evaluating physical activity levels. Therefore, this should be explicitly mentioned in the study's limitations. While the authors touch on this in Lines 356-358, I suggest specifically referencing the IPAQ and citing relevant literature, as there are many studies correlating the IPAQ with accelerometer measurements.

 Response: Thank you for your thoughtful suggestion. I agree with your advice. We added these comments in Lines 367-369.

Because the IPAQ is a retrospective questionnaire, it is subject to bias. For example, recall bias, social desirability bias, and subjective interpretation may make it difficult to assess social activity accurately.

Although not critical, I recommend that, in Table 1, the authors remove p-values and replace them with asterisks for mean or percentage values. It's important to note that specialists will be reading the paper. Additionally, for most variables, indicating one decimal place is more than sufficient; excessive precision is not required.

Response: Thank you for your thoughtful suggestion. We agree with you and have revised by removing p-values and replacing them with asterisks in Table 1.

Table 1. Baseline characteristics of the participants according to sex

Men (n = 1,030)

Women (n = 1,041)

SO† (+)
(n = 90, 8.7%)

SO (-)
(n = 940, 91.3%)

SO† (+)
(n = 108, 10.4%)

SO (-)
(n = 933, 89.6%)

Age (years)

78.7 ± 3.9**

76.1 ± 3.88

76.6 ± 3.9**

75.3 ± 3.9

Height (cm)

162.5 ± 5.8**

165.1 ± 5.5

150.3 ± 5.3**

152.1 ± 5.2

Weight (kg)

67.8 ± 9.5**

64.9 ± 8.8

58.6 ± 7.6**

56.2 ± 7.6

BMI (kg/m2)

25.6 ± 2.6**

23.7 ± 2.8

25.8 ± 2.7**

24.2 ± 2.8

Waist circumference (cm)

94.8 ± 7.8**

88.0 ± 8.2

89.7 ± 8.4**

85.6 ± 8.0

Hypertension (n, %)

60 (66.7)**

485 (51.6)

74 (68.5)*

528 (56.6)

Diabetes mellitus (n, %)

32 (35.6)**

220 (23.4)

26 (24.1)

173 (18.5)

Dyslipidemia (n, %)

27 (30.0)

223 (23.7)

52 (44.1)

373 (40.0)

Heart disease (n, %)

11 (12.2)

88 (9.4)

7 (6.5)

55 (5.9)

CVA (n, %)

4 (4.4)

56 (6.0)

4 (3.7)

28 (3.0)

Alcohol (n, %)

46 (47.8)

483 (51.4)

13 (11.6)

99 (10.6)

Current smoker (n, %)

8 (9.8)

104 (11.1)

1 (0.9)

10 (1.1)

Knee OA (n, %)

12 (13.3)

97 (10.3)

33 (30.6)

267 (28.6)

Osteoporosis (n, %)

4 (4.4)

25 (2.7)

28 (25.9)

225 (24.1)

Depression (n, %)

4 (4.4)

15 (1.6)

8 (7.4)*

26 (2.8)

MMSE-KC

26.11 ± 3.01

26.43 ± 2.67

24.4 ± 3.8**

25.4 ± 3.3

Total physical activity (METs-min/week)¶

2320 ± 1833**

4319 ± 4549

2606 ± 2242**

4158 ± 3822

Total energy expenditure (Kcal/week)§

2641 ± 2180**

4635 ± 5026

2526 ± 2224**

3851 ± 3371

Biochemical variables

Fasting glucose (mg/dL)

109.7 ± 21.7

104.8 ± 24.1

103.7 ± 30.8

102.2 ± 21.1

HbA1c (%)

6.1 ± 0.7*

5.9 ± 0.8

6.1 ± 1.1

6.03 ± 0.76

Total cholesterol (mg/dL)

169.7 ± 33.6

168.1 ± 34.9

185.0 ± 40.3

181.4 ± 35.4

HDL-C (mg/dL)

46.0 ± 13.3**

51.0 ± 14.3

57.3 ± 15.8

54.8 ± 13.5

LDL-C (mg/dL)

105.8 ± 30.3

104.1 ± 31.8

112.3 ± 35.2

113.1 ± 33.9

Protein (mg/dL)

7.0 ± 0.4

7.0 ± 0.4

6.9 ± 0.4

7.0 ± 0.3

Albumin (mg/dL)

4.3 ± 0.2

4.3 ± 0.2

4.3 ± 0.2

4.3 ± 0.2

BMI, body mass index; CVA, cerebrovascular accident; OA, osteoarthritis; MMSE-KC, Mini-Mental Status Examination-Korean version; MET, metabolic equivalent task; HDL-C, high-density lipoprotein cholesterol; LDL-C, low-density lipoprotein cholesterol.

† SO, Sarcopenic obesity: High fat mass (>29.7% for men and >37.2% for women) and low appendicular lean mass/weight measured using DXA (<29.5% for men and <23.2% for women) and either low hand grip strength (<28 kg for men and <18 kg for women) or abnormal five-time sit-to-stand chair test (≥17 s).

¶ Total physical activity (METs-min/week × min × days): (3.3[METs] × min × days) + (4.0[METs] × min × days) + (8.0(METs] × min × days).

  • Total energy expenditure (kcal/week): Total physical activity (METs-min/week)*(body weight [kg]/60)

*P < 0.05

**P < 0.01

Similarly, in Tables 2, 3, and 4, it is unnecessary to include p-values. It is well known that when the 95% CI does not contain zero, it is always statistically significant. While not obligatory, the authors could use only an asterisk to indicate whether the odds ratio is statistically significant.

Response: Thank you for your thoughtful suggestion. We agree with you and have revised by removing p-values and replacing them with asterisks in Table 2,3, and  4.

Table 2. Logistic regression analysis of physical activity levels predicting sarcopenic obesity

Unadjusted model

Fully adjusted model

Men

Women

Men

Women

Physical

activity

OR (95% CI)

OR (95% CI)

OR (95% CI)

OR (95% CI)

Moderate

activity†

0.77 (0.40–1.49)

0.83 (0.46–1.48)

0.89 (0.44–1.77)

0.92 (0.50–1.70)

High

activity††

0.47 (0.29–0.74)*

0.35 (0.22–0.55)**

0.53 (0.33–0.87)*

0.39 (0.24–0.62)**

Physical activity

category¶

8.0 MET¶¶

0.40 (0.16–0.98)*

0.39 (0.07–2.24)

0.46 (0.19–1.13)

0.56 (0.12–2.74)

4.0 MET¶¶

0.61 (0.46–0.82)*

0.77 (0.64–0.92)*

0.68 (0.51–0.91)*

0.83 (0.70–0.98)*

3.3 MET¶¶

0.73 (0.57–0.93)*

0.62 (0.47–0.82)

0.74 (0.58–0.95)*

0.66 (0.51–0.87)*

Abbreviations: OR, odds ratio; CI, confidence interval

The fully adjusted model was adjusted for age, height, hypertension, dyslipidemia, cerebrovascular accident, osteoarthritis, osteoporosis, diabetes mellitus, depression, alcohol history, smoking history, heart disease, and MMSE-KC score. Additionally, the physical activity category was adjusted for the other two intensities of physical activity.

† Low activity as reference

†† Non-high activity including low activity and moderate activity as reference

High activity, any one of the following two criteria: 7 d of walking, moderate, or vigorous intensity activities accumulating ≥3000 MET-min/week or vigorous intensity activity on ≥3 d and accumulating ≥1500 MET-min/week. Moderate activity, any of the following three criteria: ≥5 d of walking, moderate or vigorous intensity activities accumulating ≥600 MET-min/week or ≥5 d of 30 min of moderate intensity or walking per day or ≥3 d of 20 min of vigorous activity per day. Low activity is not sufficient to meet moderate or high activity criteria.

A logistic regression analysis was performed based on the hours of activity for each exercise intensity per day.

¶¶ 8.0 MET (vigorous activity), heavy lifting (>20 kg), digging or carrying things on the stairs; 4.0 MET (moderate activity), carrying light loads, parenting or cleaning; 3.3 MET (moderate activity), walking

*P < 0.05

**P < 0.01

Unadjusted model

Fully adjusted model

Men

Women

Men

Women

OR (95% CI)

OR (95% CI)

OR (95% CI)

OR (95% CI)

High fat mass¶

0.73 (0.56–0.96)*

0.68 (0.53–0.87)*

0.78 (0.59–1.04)

0.65 (0.50–0.84)*

Low muscle mass¶

0.62 (0.49–0.80)**

0.53 (0.41–0.68)**

0.67 (0.52–0.87)*

0.51 (0.39–0.66)*

Low skeletal

muscle function¶

0.55 (0.41–0.74)**

0.39 (0.28–0.52)**

0.63 (0.45–0.87)*

0.44 (0.32–0.60)*

Table 3. Logistic regression analysis of high physical activity predicting individual components of sarcopenic obesity

Abbreviations: OR, odds ratio; CI, confidence interval

The fully adjusted model was adjusted for age, height, hypertension, dyslipidemia, cerebrovascular accident, osteoarthritis, osteoporosis, diabetes mellitus, depression, alcohol history, smoking history, heart disease, and MMSE-KC score.

Non-high physical activity including low activity and moderate activity as reference

High activity, any one of the following 2 criteria: 7 days of walking, moderate or vigorous intensity activities accumulating ≥3000 MET-min/week or vigorous intensity activity on ≥3 days and accumulating ≥1500 MET-min/week. Moderate activity, any of the following 3 criteria: ≥5 days of walking, moderate or vigorous intensity activities accumulating ≥600 MET-min/week or ≥5 days of 30 min of moderate intensity or walking per day, or ≥3 days of 20 min of vigorous activity per day. Low activity is not sufficient to meet moderate or high activity criteria.

¶ High fat mass, >29.7% for men and >37.2% for women. Low muscle mass and low appendicular lean mass/weight were observed in <29.5% of men and <23.2% of women. Low skeletal muscle function, either low hand grip strength (<28 kg for men and <18 kg for women) or abnormal 5 times sit-to-stand chair test (≥17 s).

*P < 0.05

**P < 0.01

Table 4. Logistic regression analysis of high energy expenditure§ predicting sarcopenic obesity and individual components of sarcopenic obesity

Unadjusted model

Fully adjusted model

Men

Women

Men

Women

OR (95% CI)

OR (95% CI)

OR (95% CI)

OR (95% CI)

Sarcopenic Obesity†

0.39 (0.24–0.61)**

0.37 (0.24–0.57)**

0.44 (0.27–0.71)*

0.43 (0.27–0.67)*

High fat mass¶

0.88 (0.67–1.14)

0.87 (0.68–1.11)

0.90 (0.68–1.19)

0.81 (0.62–1.05)

Low muscle mass¶

0.73 (0.57–0.93)*

0.61 (0.48–0.78)**

0.76 (0.59–0.99)*

0.58 (0.44–0.75)*

Low skeletal muscle function¶

0.44 (0.33–0.59)**

0.33 (0.25–0.45)**

0.52 (0.37–0.72)*

0.40 (0.29–0.55)*

Abbreviations: OR, odds ratio; CI, confidence interval

The fully adjusted model was adjusted for age, height, hypertension, dyslipidemia, cerebrovascular accident, osteoarthritis, osteoporosis, diabetes mellitus, depression, alcohol history, smoking history, heart disease, and MMSE-KC score.

Non-high energy expenditure as reference

  • Cut-off value: 3,032 kcal/week (433 kcal/day) for men and 2,730 kcal/week (390 kcal/day) for women, calculated using the receiver operating characteristic (ROC) curve

† Sarcopenic obesity: High fat mass (>29.7% for men and >37.2% for women) and low appendicular lean mass/weight measured using DEXA (<29.5% for men and <23.2% for women) and either low hand grip strength (<28 kg for men and <18 kg for women) or abnormal five-time sit-to-stand chair test (≥17 s)

¶ High fat mass, >29.7% for men and >37.2% for women. Low muscle mass and low appendicular lean mass/weight were observed in <29.5% of men and <23.2% of women. Low skeletal muscle function, either low hand grip strength (<28 kg for men and <18 kg for women) or abnormal five-time sit-to-stand chair test (≥17 s)

*P < 0.05

**P < 0.01

Reviewer 2 Report

Comments and Suggestions for Authors

Comments on the Quality of English Language

Minor editing of English language required

Author Response

Reviewer 2

Concerning the manuscript: The Impact of Physical Activity Level on Sarcopenic Obesity

in Community-dwelling Older Adults, submitted to Healthcare. This very interesting

manuscript with relevant data in the context of Sarcopenic Obesity, but improvements can be

needed.

ï‚· The importance of study is unquestionable, but the authors do not explain the reason

(rationale) why they chose to analyze sarcopenic obesity?

Response: Thank you for your comment. We refined and added the following sentences.

(Line:57-60 )

Sarcopenia synergistically worsens the adverse effects of obesity in older adults, result-ing in sarcopenic obesity (SO). It is associated with a synergistic exacerbation effect of functional impairment, increased mortality, and reduced quality of life, leading to long-term disabilities in older adults. [10].

ï‚· Authors must be cautious in assuming that the energy expenditure - kcal/week as a synonym

for specific intensities of physical activity, as described in Line 16 (abstract) and line 66

(introduction). To avoid misinterpretations, be more specific to the terms you actually measured

(i.e. MET-min/week based on the IPAQ) and try to emphasize intensity domains/levels. The

classifications of levels of physical activity based on the IPAQ must be more explained,

referenced and even drawn (didactic figure).

Response: Thank you for your comment. We added an appendix table to avoid misinterpretations and emphasize intensity domains/levels.

Appendix 1. Physical activity category levels based on the IPAQ guidelines [22]

Physical activity category

Cut off levels

Low

no activity is reported or

not enough to meet moderate or high activity criteria

Moderate

any one of the following three criteria

≥5 days of walking, moderate or vigorous intensity activities accumulating ≥600 MET-min/week

≥5 days of 30 min of moderate intensity walking per day

≥3 days of 20 min of vigorous activity per day

High

any one of the following two criteria

7 days of walking, moderate or vigorous intensity activities accumulating ≥3000 MET-min/week

vigorous intensity activity on ≥3 days and ≥1500 MET-min/week

ï‚· Give more mathematical details about ROC analysis (in line 157 and in the figure 2).

Response: Thank you for your comment. We added more mathematical details about ROC analysis as follows: (line 260-263)

Figure 2. Receiver-operator characteristic (ROC) curve analysis of physical activity energy expenditure for sarcopenic obesity. AUC for men, 0.625 cut-off value, 3,032 kcal/week (54.0% sensitivity and 68.9% specificity, (95% CI, 0.57–0.68); AUC for women, 0.643 cut-off value, 2,730 kcal/week (53.3% sensitivity and 70.4% specificity (95% CI, 0.59–0.70)

ï‚· About logistic regression analysis: odds ratio of low activity and moderate activity are not

being displayed (Table 3). The same for the table 4 since odds ratio of non-high energy

expenditure group are not being displayed.

Response: Thank you for your comment. Because this table shows the OR for non-high physical activity compared to the high physical activity group, the odds ratio of low activity and moderate activity are not displayed.

ï‚· About logistic regression analysis and (ROC) curve analysis. I would like to know the authors' line of reasoning about the usefulness of these analyses in the science (in discussion).

Response: Thank you for your comment.  We suggested the physical activity categories and the total physical energy expenditures of kcal/week both are all meaningful. Its application in patients could be selected by the researcher. Thank you.

ï‚· Give details about the procedures. It is important to describe when data were collected (time

of day, environmental condition, season of data collection, data collection duration). Were

researchers trained or had training in conducting tests?

Response: Thank you for your comment. We added more details of Korean frailty and aging cohort study (KFACS): cohort profile. (Line 85-87)

ï‚· Table 1 could be more explored by comparing groups with effect size (Cohen's d) and

percentage differences. The P-value should continue because it's great.

Response:  Thank you for your advice. We will apply it in further research.

ï‚· The authors must insert more comparisons. For example, table 1 (according to sex) is great,

but another table mixing data from men and women (according to SO†) would be also

interesting.

Response:  Thank you for your advice. Because the diagnostic criteria for SO are different for men and women, it is compared separately. Thank you.

ï‚· Regarding data exploration, I suggest the inclusion of Pearson’s correlations for

understanding interactions. This could be made using data from participants of all groups (for

amplifying the sample amount). If using figures, I suggest that you accurately discriminate the

groups using different symbols or colors. For example:

men - SO† > square red

men - SO (-) > square green

woman - SO† > circle blue

woman - SO (-) > circle yellow

However, it is necessary to reexamine the research method etc. in several respects.

Response: Thank you for your comment. This is a very interesting advice. The goal of this study was to find Physical activity category levels and cut-off values indicating that total physical activity energy expenditure of > 3,032 kcal/week (433 kcal/day) for men and 2,730 kcal/week (390 kcal/day) affects SO incidence. We will apply Pearsons correlations in subsequent research. Thank you.

Reviewer 3 Report

Comments and Suggestions for Authors

The aim of this study was to assess the impact of the intensity of physical activity on the risk of sarcopenic obesity (SO). To investigate this, cross-sectional data from2071 older adults were analysed. It was seen in multiple regression analysis that moderate-intensity exercise and an energy expenditure >3032 kcal/wk or 2730 kcal/wk for men and women, respectively were associated with a lower prevalence of SO.

Overall, a well-written piece of work.

Minor comments:

Perhaps in the abstract in line 256 replace ‘risk’ with ‘prevalence’ as that is what you determined in your work.

Introduction line 35-36 can you perhaps specify that it is reduced physical activity levels, and perhaps concerning eating habits specify how the eating habits change. Same for the hormones. Using words like ‘change, difference, altered etc’ leaves questions on how they the are ‘changed, differ, altered’.

Line 40-41 ‘…fat accumulation due to weight gain…’ is an odd construct. The fat is not accumulating due to weight gain, but rather the fat accumulation contributes (perhaps is even the sole cause) of weight gain.

Line 41-43: It is not necessarily that these conditions follow weight gain, but rather there is an increased risk developing these conditions.

Line 44: Remove ‘in advance’

Page 2 line 50: Perhaps it is due to ageing, but there are master athletes who remain very active in old age. I suggest replacing ‘due to’ with ‘often seen during’. Also, the reduced metabolic rate does not contribute to a decrease in muscle mass, but rather is perhaps caused by the reduction in muscle mass.

Page 2 line 54: The lower testosterone is not so much ‘affecting muscle loss’ but rather ‘may contribute to the loss of muscle mass’.

Page 2 line 55: Estrogen does not modulate inflammation through satellite cell activation. Please rephrase.

Page 2 line 62-63: this is not ‘resulting in SO’, but rather ‘increasing the risk of developing SO’.

Page 5 lines 172-175: For readability don’t repeat means ± SD that are given in the table.

In table1, I suggest to have not more than 3 digits per parameter. It is better to say 163 cm than 162.55 cm for instance.

Make a clearer distinction between text and legends.

Page 9 line 290: Replace ‘lower risk’ with ‘lower prevalence’. This applies also throughout the Discussion in most cases.

Author Response

Reviewer 3

The aim of this study was to assess the impact of the intensity of physical activity on the risk of sarcopenic obesity (SO). To investigate this, cross-sectional data from 2071 older adults were analysed. It was seen in multiple regression analysis that moderate-intensity exercise and an energy expenditure >3032 kcal/wk or 2730 kcal/wk for men and women, respectively were associated with a lower prevalence of SO.

Overall, a well-written piece of work.

Minor comments:

Perhaps in the abstract in line 256 replace ‘risk’ with ‘prevalence’ as that is what you determined in your work.

Response: Thanks for your advice. Since our research results look at the prevalence, replace ‘risk’ with ‘prevalence.’ This has become an object throughout the article. Thank you.

Introduction line 35-36 can you perhaps specify that it is reduced physical activity levels, and perhaps concerning eating habits specify how the eating habits change. Same for the hormones. Using words like ‘change, difference, altered etc’ leaves questions on how they the are ‘changed, differ, altered’.

Response: Thanks for your advice. For clarity, the sentence was changed as follows:

( LINE 35-36)

Obesity in older adults is related to lifestyle changes, including low physical activity, high calories eating habits, and physiological changes such as lower sex hormones

Line 40-41 ‘…fat accumulation due to weight gain…’ is an odd construct. The fat is not accumulating due to weight gain, but rather the fat accumulation contributes (perhaps is even the sole cause) of weight gain.

Response: Thanks for your advice. For clarity, the sentence was changed as follows:

(LINE 40-42)

Obesity causes metabolic syndrome in older adults and weight gain due to fat accumula-tion induces insulin resistance and hyperinsulinemia in the liver and peripheral tissues.

Line 41-43: It is not necessarily that these conditions follow weight gain, but rather there is an increased risk developing these conditions.

Response: Thanks for your advice. For clarity, the sentence was changed as follows: (LINE 42-44)

This could increase the risk of developing hypertension, glucose intolerance, diabetes, visceral obesity, dyslipidemia, cardiovascular disease, and some cancers

Line 44: Remove ‘in advance’

Response: Thanks for your advice. We removed it.

Page 2 line 50: Perhaps it is due to ageing, but there are master athletes who remain very active in old age. I suggest replacing ‘due to’ with ‘often seen during’. Also, the reduced metabolic rate does not contribute to a decrease in muscle mass, but rather is perhaps caused by the reduction in muscle mass.

Response: Thanks for your advice. For clarity, the sentence was changed as follows: (LINE 49-51)

Decreased total energy consumption, such as reduced resting metabolic rate and lower physical activity often seen during aging, contributes substantially to increase in body fat mass [6].

Page 2 line 54: The lower testosterone is not so much ‘affecting muscle loss’ but rather ‘may contribute to the loss of muscle mass’.

Response: Thanks for your advice. For clarity, the sentence was changed as follows:

(LINE 53-54)

However, testosterone levels decrease by 1–3% annually with age, may contribute to mus-cle loss and fat distribution in older men

Page 2 line 55: Estrogen does not modulate inflammation through satellite cell activation. Please rephrase.

Response: Thanks for your advice. For clarity, the sentence was changed as follows: (LINE 54-55)

In women, estrogen modulates inflammation, and satellite cell activation in skeletal muscles.

Page 2 line 62-63: this is not ‘resulting in SO’, but rather ‘increasing the risk of developing SO’.

Response: Thanks for your advice. For clarity, the sentence was changed as follows:

(LINE 61-63)

Physical activity decreases with age, and reports have suggested that the degree of physical activity in daily life affects the prevalence of sarcopenia and obesity, increasing the risk of developing SO [11].

Page 5 lines 172-175: For readability don’t repeat means ± SD that are given in the table.

Response: Thanks for your advice. We deleted repeated means ± SD.

In table1, I suggest to have not more than 3 digits per parameter. It is better to say 163 cm than 162.55 cm for instance.

Response: Thanks for your advice. We changed them to 3 digits per parameter throughout the Tables.

Make a clearer distinction between text and legends.

Response: Thanks for your advice. A blank line has been added so that it will be clearly distinguished during the publishing process.

Page 9 line 290: Replace ‘lower risk’ with ‘lower prevalence’. This applies also throughout the Discussion in most cases. Concerning the manuscript: The Impact of Physical Activity Level on Sarcopenic Obesity in Community-dwelling Older Adults, submitted to Healthcare. This very interesting manuscript with relevant data in the context of Sarcopenic Obesity, but improvements can be needed.

Response: Thanks for your advice. Since our research results look at the prevalence, replace ‘risk’ with ‘prevalence.’ This has become an object throughout the article. Thank you.